# Synthesis, Crystal Structure and Bioactivity of Phenazine-1-carboxylic Acylhydrazone Derivatives

**DOI:** 10.3390/molecules26175320

**Published:** 2021-09-01

**Authors:** Shouting Wu, Xi Liang, Fang Luo, Hua Liu, Lingyi Shen, Xianjiong Yang, Yali Huang, Hong Xu, Ning Wu, Qilong Zhang, Carl Redshaw

**Affiliations:** 1Research Center for Molecular Medical Engineering, Department of Basic Medical Sciences, Guizhou Medical University, Guiyang 550004, China; wushouting1996@126.com (S.W.); liangxi_15@163.com (X.L.); shen-ly@stumail.nwu.edu.cn (L.S.); yangxianjiong@126.com (X.Y.); ylh6401@gmc.edu.cn (Y.H.); xuhong@gmc.edu.cn (H.X.); 2Department of Biology & Engineering, Guizhou Medical University, Guiyang 550004, China; luo2020110110128@126.com (F.L.); liuhuaa008@126.com (H.L.); 3Department of Chemistry, University of Hull, Cottingham Road, Hull HU6 7RX, UK; c.redshaw@hull.ac.uk

**Keywords:** phenazine-1-carboxylic acid, acylhydrazone compounds, crystal structure, anticancer activity

## Abstract

A phenazine-1-carboxylic acid intermediate was synthesized from the reaction of aniline and 2-bromo-3-nitro-benzoic acid. It was then esterified and reacted with hydrazine hydrate to afford phenazine-1-carboxylic hydrazine. Finally, 10 new hydrazone compounds **3a**–**3j** were obtained by the condensation reaction of phenazine-1-carboxylic acid hydrazide and the respective aldehyde-containing compound. The structures were characterized by ^1^H and ^13^C NMR spectroscopy, MS and single crystal X-ray diffraction. The antitumor activity of the target compounds in vitro (HeLa and A549) was determined by thiazolyl blue tetrazolium bromide. The results showed that compound (*E*)-*N*′-(2-hydroxy-4-(2-(piperidine-1-yl) ethoxy) benzyl) phenazine-1-carbonyl hydrazide **3d** exhibited good cytotoxic activity.

## 1. Introduction

Acylhydrazone is formed by the condensation of hydrazine and an aldehyde or ketone, containing both oxygen and nitrogen atoms that are involved in the formation of hydrogen bonds in living organisms and increase the affinity between receptors, thereby, inhibiting numerous physiological and chemical processes in vivo [1,2,3,4,5]. Acylhydrazone compounds are special in terms of both their structure and properties and have been the focus of considerable research by chemists. The -CONH-N=CH- group has a large conjugate system, exhibits favorable stability and coordination ability [6,7,8], and can readily form chelates with iron/ferrous ions in the body. By changing the form of iron in this way, it prevents the absorption and utilization of iron by cells, thereby changing the distribution of iron in the body. In other words, it can regulate the distribution and metabolism of iron in the body. In particular, it can reduce cancer cell and tumor tissue iron content or can reduce the effective use of iron by cancer cells. It has the ability to make cancer cells and tissues iron “hungry” and can affect the activity of enzymes containing iron. In this way, the cancer cells′ energy metabolism, DNA replication, and transcription as well as protein synthesis and folding can be retarded or inhibited. The importance is highlighted because the division and growth can lead to the death [9,10,11,12,13]. Therefore, many hydrazone compounds are potential anticancer drugs because of their good anticancer activity [14].

Phenazine-1-carboxylic acid (PCA) is an antibiotic that can be used to protect crops against a broad spectrum of soil-borne fungal *Pseudomonas* spp. [15]. It is widely found in the metabolites of many microorganisms, such as streptomyces [16] and Pseudomonads [17]. PCA can be isolated from the fermentation broth of Pseudomonas M-18 [18]. It was found to have antituberculous bacterial properties and can inhibit seaweed growth [19,20]. PCA was confirmed by the Chinese pesticide naming unit, named as *Shenazinamycin*, which is an antibiotic secreted by *Pseudomonas fluoresceae* through biological culture. It has the dual functions of a broad-spectrum of inhibition of plant pathogens and the promotion of plant growth [21]. Rewcastle et al. synthesized a series of PCA analogs on the basis of original natural products and developed lead compounds possessing antilung cancer [22] and leukemia [23] properties and is currently in Phase II clinical trials [24]. PCA and its analogs isolated from secondary metabolites of *Streptomyces* sp. IFM 11694 can overcome the resistance of human gastric adenocarcinoma cells to TRAIL, a tumor necrosis factor related apoptotic inducing ligand, showing good antitumor activity [25]. However, the synthesis and bioactivity of PCA based acylhydrazones have not yet been reported. Therefore, we first synthesized PCA by a chemical synthesis method and then synthesized 10 new hydrazone compounds **3a**–**3j** (Scheme 1) based on PCA. We also investigated the antitumor activity of the target compounds in vitro. Chemical synthesis of phenazine natural products by means of structural modification or new synthesis of phenazine products are also emerging. The antitumor activity and structure activity relationship are also attracting increasing attention, and the mechanism of action may be related to interfering with respiratory chain, inhibiting topoisomerase activity and DNA breaking [26,27].

## 2. Results and Discussion

### 2.1. Synthesis and Crystal Structures of the Target Compounds

Phenazine biosynthesis has attracted attention since its early studies in the second half of the 20th century [28]. Ever since the total synthesis of PCA was reported by Kogl and Postowsky [29] in 1930, new reactions and synthetic strategies have emerged over the following half century, and the yield and route have been continuously improved and optimized. The synthetic route to the target compounds **3a**–**3j** is shown in Scheme 1, and the structure and yields are shown in Table 1. According to reference [30], PCA was synthesized without the need for column chromatography during the synthesis process. Instead, relatively pure PCA was obtained by recrystallization from acetonitrile and ethanol (1:1). Then, PCA was esterified in concentrated sulfuric acid, and reacted with hydrazine hydrate to form acylhydrazine. Finally, the acylhydrazine compounds were condensed with aldehyde-containing compounds to afford the target compounds **3a**–**3j**. **3a**–**3j** were dissolved in the mixed solvent system of chloroform and ethanol; the ratio of chloroform to ethanol is 1:1. Heating with frequent agitation, boiling to effect solution, and then slowing evaporation at room temperature resulted in 9 sets of crystals suitable for X-ray diffraction. The single crystal X-ray diffraction was performed, and the crystal structures revealed that the aldehyde compounds had reacted successfully with the amino group of the hydrazide. In all crystal structures, the hydrogen on the amide nitrogen atom and the nitrogen atom on the phenazine ring formed intramolecular N-H...N hydrogen bonds. The aldehydes containing ortho hydroxyl groups reacted with acylhydrazine to afford acylhydrazone compounds. Hydrogen atoms of the ortho hydroxyl groups and the imine nitrogen atoms also formed O-H...N intramolecular hydrogen bonds; thus, both the hydroxyl and carbonyl groups are on the same side, but on the other side of the phenazine ring (See Figure 1 for an example of the crystal structure of **3b**; hydrogen bond distances are shown in Appendix A).

### 2.2. Anticancer Activity of ***3a**–**3f*** in Hela and A549 Cell Lines

In our study, results are compared with cisplatin (20 μmol/L). Hela and A549 cells were seeded onto 96-well plates at the proper density and treated with **3a**–**3j** at final concentrations for 24 h; cell viability was analyzed via an MTT assay. There was a pool of three different sets of experiments, each repeated in triplicate. Statistical comparisons were conducted. The results are shown in Table 2.

The activity showed that the compounds **3b**–**3f** were effective in the Hela and A549 cell lines, among which **3d** had the strongest effect on Hela and A549 cells. Compound **3d** revealed better cytotoxic activity than cisplatin, with IC50 values of 5.5 ± 1.3 and 2.8 ± 1.53 μmol/L against HeLa and A549, respectively, with statistical significance (*** *p* < 0.001, significantly different from the value of control group, *n* = 3). 

The antitumor activity of phenazine compounds has been widely studied, and can induce cell apoptosis and cycle arrest. The mechanism of action is related to the inhibition of topoisomerase and the induction of reactive oxygen species. DNA, which is regarded as an important genetic material of cell life, has become the target of many antitumor drugs. Phenazine compounds have planar macromolecular structures for intramolecular hydrogen bonding. The shorter the hydrogen bond distance, the more stable the structure. It may bind to DNA in a way similar to ethidium bromide and campinetin [31].

According to previous reports, compounds that bear hydrazide-hydrazone moiety have received considerable attention due to their biological value in the development of novel antimicrobials and analgesic, anti-inflammatory, and anticancer agents [32]. The compound PAC-1 (Figure 2) was reported to enhance the activity of procaspase-3 in vitro and induce apoptosis in tumor cells in vivo [33]. The cells were treated with different concentrations (0, 0.1, 1, 10, 100, and 1000 μmol/L) of synthesized compounds. Picolinoylhydrazone derivative (**1**) (Figure 3) is a potent growth inhibitor of lung, leukemia and ovarian cancer cell lines, with IC50 values below 10 μmol/L [34]. A bis-isatin hydrazones of imidazolidine-2, 4-dione (**2**) (Figure 4) showed good antiproliferative activity against the human colon cancer cell line HCT-116.

A series of novel pyrano[3,2-a]phenazine derivatives (Figure 5), designed as hybrid molecules of phenazine and pyran pharmocophores and then treated with compounds at concentrations from 0 to 50 μmol/L. Cytotoxic evaluation indicates that many compounds exhibited cytotoxicity against HCT116, MCF7, HepG2 and A549 cancer cell lines in vitro, in which compounds **1c**, **1i**, **2e**, and **2l** were found to have excellent antiproliferative activity against the HepG2 cancer cell line [35]. This provided the experimental basis for the further mechanism of **3a**–**3j** and its influence on tumor cells.

Compared with the above structures, **3a**–**3j** all have a hydrazine structure and IC50 values of **3d** below 10 μmol/L. Significantly, **3d** showed more potency than the positive control drug, which is a potent growth inhibitor of HeLa and A549 cells. It can be seen from Table 2 that the hydrazine obtained by the condensation reaction of phenazine-1-hydrazine with the salicylaldehyde and salicylaldehyde analogs have certain antitumor activity, while the hydrazine obtained by the condensation reaction of other aldehydes with phenazine-1-hydrazine has no activity. The specific antitumor activity mechanism, whether by inhibiting enzyme activity or not, needs further study.

## 3. Materials and Methods

### 3.1. Instruments and Reagents

^1^H/^13^C NMR spectra were measured on a Inova-400-Bruker AV600 NMR spectrometer, with *d*-DMSO as solvent and tetramethylsilane as internal reference (600 MHz), and the *J* value was in Hz. High resolution mass spectrometry (HRMS) was performed using a GCT premier CAB048 mass spectrometer. Data for the crystal structures were collected on a Bruker Smart Apex single crystal diffractometer. Melting points were measured on a Yanaco microscopic melting point meter, and the temperature was not corrected. Hela and A549 cells were replaced in a fresh culture medium and incubated for 24 h in a humidified atmosphere of 95% air and 5% CO_2_ at 37 °C (Thermo-Fisher, New York, NY, USA). MTT assay was analyzed by Epoch Continuous Wavelength Microplate (Bio-Tek, Burlington, VT, USA) and RPMI-1640 medium (Gibco, New York, NY, USA). Solvents and reagents were obtained commercially and were of analytical grade.

### 3.2. Determination of the Crystal Structures of the Compounds

The target compounds were dissolved in chloroform and slowly evaporated for 5–10 days to obtain crystals of appropriate size. A Bruker Smart Apex CCD single crystal diffractometer and a graphite monochromator with monochromatized Mo *Kα* rays (*λ* = 0.071073 nm) were used to collect single crystal diffraction data within a certain θ range by *φ-ω* scanning mode. The diffraction intensity data were corrected by empirical absorption and LP. The crystal structures were solved by direct methods. All non-hydrogen coordinates and their anisotropic thermal parameters were modified by the full matrix least square method. All calculations were completed with the Shelx-97 program [36]. Crystallographic data and refinement details for **3a**–**3j** are given in Appendix A. CCDC: 2102942, **3a**. CCDC: 2102943, **3b**. CCDC: 2102944, **3c**. CCDC: 2102945, **3e**. CCDC: 2102946, **3f**. CCDC: 2102947, **3g**. CCDC: 2102948, **3h**. CCDC: 2102949, **3i**. CCDC: 2102950, **3j**. These data can be obtained free of charge from the Cambridge Crystallographic Data Centre www.ccdc.cam.ac.uk/data_request/cif (accessed on 13 August 2021).

### 3.3. Synthesis of Compounds

#### 3.3.1. Synthesis of PCA

Aniline (2.5 g, 26.8 mmol), 2-bromo-3-nitrobenzoic acid (6.3 g, 26.6 mmol), CuI (1.3 g, 6.8 mmol) and triethylamine (12.5 mL, 90.0 mmol) were weighed out and combined with glycol (50.0 mL, 899.1 mmol), dissolved in a 100 mL three-mouth flask and the system heated at 95 °C for 3 h. On cooling to room temperature, it was poured into a 0.2 mol/L sodium hydroxide (300 mL, 60 mmol) solution. Work-up included silica gel adsorption decolorization and extraction and filtration to afford a red solution, which was then acidified with dilute hydrochloric acid to pH = 3, affording the product 3-nitro-2-(phenylamino) benzoic acid. This product was pure enough to be directly used in the next reaction.

The solution of 3-nitro-2-(phenylamino) benzoic acid (4.0 g, 15.5 mmol) and NaBH_4_ (4.8 g, 126.0 mmol) was dissolved in 2 mol/L sodium hydroxide (450 mL, 900 mmol), refluxed for 4 h, and then the sodium salt solution of phenazine carboxylic acid was obtained after cooling. A green solid was obtained by adjusting the pH of the solution with dilute hydrochloric acid to pH = 3. The solid was recrystallized with a mixture of acetonitrile and ethanol (1:1) to obtain yellow-green crystalline PCA.

#### 3.3.2. Synthesis of Phenazine-1-hydrazide

Concentrated sulfuric acid (1 mL) was slowly added to 100 mL of anhydrous ethanol in an ice bath, restored to room temperature, and then PCA (2.24 g, 10 mmol) was added to the solution, which was refluxed for 21 h. Then the solution was cooled to room temperature. The concentrated solution was steamed to 50 mL, and the solution was poured into 200 mL of ice water. With saturated potash, the pH was adjusted to pH = 10, and then extraction with dichloromethane (60 mL × 4) was conducted. The organic phase was collected and combined, and then dried overnight with anhydrous sodium sulfate and filtered and rotary steamed to remove the dichloromethane to afford a black solution. Then, 40 mL anhydrous ethanol and 5 mL hydrazine hydrate were added to the black solution, and the system was refluxed for 8 h. After cooling, the solids were separated out and filtered, and phenazine-1-hydrazine was obtained after washing with anhydrous ethanol for three times and drying.

#### 3.3.3. Synthesis of the Acylhydrazone Compounds

Ten aldehydes were weighed out, and each was dissolved in 30 mL anhydrous ethanol with phenazine-1-hydrazine at a molar ratio of 1:1. Following refluxing for 8 h and upon cooling, a solid separated out, and filtration followed by ethanol washing for 3 times to obtain a brown solid. The solid was recrystallized with chloroform and ethanol and filtered and dried to obtain the acylhydrazone compounds.

*(E)-N′-(isoquinoline-3-methyl)phenazine-1-carbonyl hydrazide* (**3a**): brown solid 0.23 g, yield 60%, m. p. 256–257 °C; ^1^H NMR (600 MHz, *d*-DMSO): δ_H_ (ppm): 13.28 (s, 1H, N**H**), δ 8.72 (s, 1H, isoquinoline-**H**), δ 8.60 (d, *J* = 6.0 Hz, 1H, quinoline-**H**), 8.53 (d, *J* = 12.0 Hz, 1H, isoquinoxaline-**H**), 8.5. (d, *J* = 6.0 Hz, 1H, quinoline-**H**), 8.48 (d, *J* = 6.0 Hz, 1H, isoquinoxaline-H), 8.34 (t, *J* = 12.0 Hz, 1H, isoquinoline-**H**), 8.23 (s, 1H, N=C**H**), 8.14–8.08 (d, *J* = 6.0 Hz, 2H, quinoline-H), 8.7–8.04 (m, 2H, quinoline-**H**), 7.92–7.82 (m, 2H, isoquinoline-**H**), 7.68 (d, *J* = 12.0 Hz, 1H, isoquinoline-**H**). ^13^C NMR (150 MHz, *d*-DMSO): δ_C_ (ppm): 162.0, 153.7, 148.4, 147.3, 142.8, 142.5, 141.6, 139.8, 136.7, 133.3, 132.8, 132.0, 131.8, 131.5, 130.3, 130.0, 129.6, 129.1, 128.8, 128.0, 127.9, 127.3, 117.6. HRMS (ESI+) *m*/*z* Calcd for C_23_H_15_N_5_O [M + H]^+^ 378.1322, found: 378.1352.

*(E)-N′-(2-hydroxybenzyl)phenazine-1-carbonyl hydrazide* (**3b**): brownish yellow powder 0.22 g, yield 65%, m. p. 270–271 °C; ^1^H NMR (600 MHz, *d*-DMSO): δ_H_ (ppm): 13.28 (s, 1H, N**H**), 11.36 (s, 1H, O**H**), 8.82 (s, 1H, N=C**H**), 8.63 (d, *J* = 6.0, 1H, quinoxaline-**H**), 8.57–8.47 (d, *J* = 12.0 Hz, 2H, quinoxaline-**H**), 8.34 (d, *J* = 12.0 Hz, 1H, quinoxaline-**H**) 8.16–8.06 (m, 3H, quinoxaline-**H**), 7.64 (d, *J* = 6.0 Hz, 1H, benzylidenimin-**H**), 7.36 (t, *J* = 6.0 Hz, 1H, benzylidenimin-**H**), 7.00 (d, *J* = 6.0 Hz, 1H, benzylidenimin-**H**), 6.99 (t, *J* = 6.0 Hz, 1H, benzylidenimin-**H**). ^13^C NMR (150 MHz, *d*-DMSO): δ_C_ (ppm): 161.3, 157.5, 149.3, 142.8, 142.5, 141.5, 139.7, 133.6, 132.9, 132.0, 131.8, 131.5, 130.7, 130.3, 129.8, 129.4, 129.2, 119.3, 118.5, 116.4. HRMS (ESI+) *m*/*z* Calcd for C_20_H_14_N_4_O_2_ [M + H]^+^ 343.1211, found: 343.1195.

*(E)-N′-(3,5-dibromo-2-hydroxybenzy)phenazine-1-carbonyl hydrazide* (**3c**): yellow powder 0.32 g, yield 65%, m. p. 269–270 °C; ^1^H NMR (600 MHz, *d*-DMSO): δ_H_ (ppm): 13.70 (s, 1H, N**H**), 12.64 (s, 1H, O**H**), 8.80 (s, 1H, N=C**H**), 8.67 (d, *J* = 6.0 Hz, 1H, quinoxaline-**H**), 8.54 (d, *J* = 12.0 Hz, 2H, quinoxaline-**H**), 8.36 (d, *J* = 6.0 Hz, 1H, quinoxaline-**H**), 8.18–8.06 (m, 3H, quinoxaline-**H**), 7.88 (s, 2H, benzylidenimin-**H**). Due to the poor solubility of compound **3c**, we could not obtain ^13^C-spectral data, HRMS (ESI+) *m*/*z* Calcd for C_20_H_12_Br_2_N_4_O_2_ [M + H]^+^ 498.9310, found: 498.9400.

*(E)-N′-(2-hydroxy-4-(2-(piperidine-1-yl)ethoxy)benzyl) phenazine-1-carbonyl hydrazide* (**3d**): orange yellow powder 0.26 g, yield was 55%, m. p. 203–204 °C; ^1^H NMR (600 MHz, *d*-DMSO): δ_H_ (ppm): 13.24 (s, 1H, N**H**), 11.66 (s, 1H, O**H**), 8.73 (s, 1H, N=C**H**), 8.63 (d, *J* = 6.0 Hz, 1H, quinoxaline-**H**), 8.55–8.45 (d, *J* = 6.0 Hz, 2H, quinoxaline-C**H**), 8.34 (d, *J* = 6.0 Hz, 1H, quinoxaline-**H**), 8.18–7.92 (m, 3H, quinoxaline-**H**), 7.50 (s, 1H, benzylidenimin-**H**), 6.58 (s, 1H, benzylidenimin-**H**), 6.54 (d, *J* = 6.0 Hz, 1H, benzylidenimin-**H**), 4.12–2.62 (t, *J* = 6.0 Hz, 4H, methylene-**H**), 2,48–1.39 (m, 10H, piperidine-**H**). ^13^C NMR (150 MHz, *d*-DMSO): δ_C_ (ppm): 161.4, 160.8, 159.5, 149.8, 142.7, 142.6, 141.4, 139.7, 133.6, 132.9, 132.0, 131.8, 131.4, 130.5, 130.3, 129.4, 129.1, 111.5, 107.6, 106.9, 101.6, 101.2, 65.7, 57.2, 54.3, 25.5, 223.8. HR-MS (ESI+) *m*/*z* Calcd for C_27_H_27_N_5_O_3_ [M]^+^ 470.2102, found: 470.2194.

*(E)-N′-(2-hydroxy-3-methoxybenzyl)phenazine-1-carbonyl hydrazide* (**3e**): brown solid 0.25 g, yield 68%, m. p. 190–191 °C; ^1^H NMR(600 MHz, *d*-DMSO): δ_H_ (ppm): 13.24 (s, 1H, O**H**), 11.10 (s, 1H, N**H**), 8.81 (s, 1H, N=C**H**), 8.63–8.26 (d, *J* = 12.0 Hz, 4H, quinoxaline-**H**), 8.19–7.92 (m, 3H, quinoxaline-**H**), 7.25–7.08 (d, *J* = 12.0 Hz, 2H, benzylidenimin-**H**), 6.92 (t, *J* = 12.0 Hz, 1H, benzylidenimin-C**H**), 3.85 (s, 3H, OC**H**_3_). ^13^C NMR (150 MHz, CHCl_3_): δ_C_ (ppm): 161.2, 149.1, 147.8, 147.2, 142.7, 142.5, 141.4, 139.7, 133.6, 132.9, 132.0, 131.7, 130.6, 129.4, 129.1, 121.1, 118.9, 118.6, 113.9, 79.0, 55.7. HRMS (ESI+) *m*/*z* Calcd for C_21_H_16_N_4_O_3_ [M + H]^+^ 373.1102, found: 373.1298.

*(E)-N′-(2,4-dihydroxybenzyl)phenazine-1-carbon hydrazide* (**3f**): yellow-brown powder 0.25 g, the yield of 70%, m. p. 297–298 °C; ^1^H NMR (600 MHz, *d*-DMSO): δ_H_ (ppm): 13.18 (s, 1H, O**H**), 11.54 (s, 1H, O**H**), 10.04 (s, 1H, N**H**), 8.67 (s, 1H, N=C**H**), 8.61 (d, *J* = 6.0 Hz, 1H, quinoxaline-**H**), 8.50-8.27 (d, *J* = 12.0 Hz, 3H, quinoxaline-**H**), 8.09-8.02 (m, 3H, quinoxaline-**H**), 7.40 (d, *J* = 6.0 Hz, 1H, benzylidenimin-**H**), 6.42 (s, 1H, benzylidenimin-**H**), 6.37 (d, *J* = 6 Hz, 1H, benzylidenimin-**H**). ^13^C NMR (150 MHz, *d*-DMSO): δ_C_ (ppm):160.8, 160.6, 159.6, 150.1, 142.7, 142.5, 141.2, 139.6, 133.7, 132.9, 132.0, 131.7, 131.7, 130.3, 130.3, 129.3, 129.1, 110.4, 107.7, 102.6. HRMS (ESI+) *m*/*z* Calcd for C_20_H_14_N_4_O_3_ [M + H]^+^ 359.1102, found: 359.1140.

*(E)-N′-(pyridine-2-methyl)phenazine-1-carbonyl hydrazide* (**3g**): reddish-brown crystalline 0.20 g, yield 60%, m. p. 220–221 °C; ^1^H NMR (600 MHz, *d*-DMSO): δ_H_ (ppm): 13.10 (s, 1H, N**H**), 8.67 (d, *J* = 6.0 Hz, 1H, 2-pyridine-**H**), 8.56–8.49 (d, *J* = 6.0 Hz, 2H, quinoxaline-**H**), 8.48 (s, 1H, N=C**H**), 8.33 (t, *J* = 12.0 Hz, 1H, quinoxaline-**H**), 8.12 (d, *J* = 6.0 Hz, 1H, 2-pyridine-**H**), 8.07 (t, *J* = 6.0 Hz, 1H, quinoxaline-**H**), 8.06–8.02 (m, 2H, quinoxaline-**H**), 7.95 (t, *J* = 12.0 Hz, 2H, 2-pyridine-**H** ), 7.47 (s, 1H, N=C**H**). ^13^C NMR (150 MHz, *d*-DMSO): δ_C_ (ppm): 161.9, 153.1, 149.5, 148.4, 142.8, 142.5, 141.6, 139.8, 136.8, 133.0, 132.7, 132.0, 131.8, 131.7, 130.3, 129.5, 129.1, 124.5, 120.7. HRMS (ESI+) *m*/*z* Calcd for C_19_H_13_N_5_O [M + H]^+^ 328.1102, found: 328.1196.

*(E)-N′-(pyridine-3-methyl)phenazine-1-carbonyl hydrazide* (**3h**): yellowish brown crystal 0.20 g, yield 60%, m. p. 230–231 °C; ^1^H NMR (600 MHz, *d*-DMSO): δ_H_ (ppm): 13.37 (s, 1H, N**H**), 8.97 (s, 1H, 3-pyridine-**H**), 8.67 (s, 1H, N=C**H**), 8.64 (d, *J* = 6.0 Hz, 1H, 3-pyridine-**H**), 8.53 (d, *J* = 6.0 Hz, 1H, 3-pyridine-**H**), 8.48 (d, *J* = 6.0 Hz, 1H, quinoxaline-**H**) 8.34–8.24 (d, *J* = 6.0 Hz, 2H, quinoxaline-**H**), 8.11 (d, *J* = 6.0 Hz, 1H, 3-pyridine-**H**), 8.11–8.03 (m, 3H, quinoxaline-**H**), 7.55 (t, *J* = 6.0 Hz, 1H, 3-pyridine-**H**). ^13^C NMR (150 MHz, *d*-DMSO): δ_C_ (ppm): 161,4, 150.7, 148.8, 145.8, 142.7, 142.6, 141.3, 139.7, 133.7, 133.5, 133.0, 132.0, 131.8, 130.7, 130.3, 130.2, 129.4, 129.2, 124.0. HRMS (ESI+) *m*/*z* Calcd for C_19_H_13_N_5_O [M + H]^+^ 328.1102, found: 328.1200.

*(E)-N′-(pyridine-4-methyl)phenazine-1-carbonyl hydrazide* (**3i**): yellow powder 0.20 g, yield 60%, m. p. 243–244 °C; ^1^H NMR (600 MHz, *d*-DMSO): δ_H_ (ppm): 13.42 (s, 1H, N**H**), 8.72 (d, *J* = 6.0 Hz, 2H, 4-pyridine-**H**), 8.62 (s, 1H, N=C**H**), 8.57–8.47 (d, *J* = 6.0 Hz, 2H, quinoxaline-**H**), 8.37–8.32 (d, 2H, quinoxaline-**H**), 8.14–8.05 (m, 3H, quinoxaline-**H**), 7.78 (d, *J* = 6.0 Hz, 2H, 4-pyridine-**H**). ^13^ C NMR (150 MHz, CHCl_3_): δ_C_ (ppm): 161.3, 151.1, 148.5, 143.4, 141.0, 140.2, 136.6, 134.6, 132.3, 131.4, 130.2, 130.0, 128.6, 127.8, 122.6, 119.0, 118.0, 114.1, 56.3. HRMS (ESI+) *m*/*z* Calcd for C_19_H_13_N_5_O [M + H]^+^ 328.1102, found: 328.1198.

*(E)-N′-((2-hydroxynaphthalene-1-yl)methyl)phenazine-1-carbonyl hydrazide* (**3j**): orange powder 0.24 g, yield 62%, m. p. 276–277 °C; ^1^H NMR (600 MHz, *d*-DMSO): δ_H_ (ppm): 13.45 (s, 1H, O**H**), 13.01 (s, 1H, N**H**), 9.67 (s, 1H, N=C**H**), 8.72 (d, *J* = 6.0 Hz, 1H, quinoxaline-**H**), 8.69 (d, *J* = 6.0 Hz, 1H, 1-naphthalene-**H**), 8.52–8.32 (d, *J* = 12 Hz, 3H, quinoxaline-**H**), 8.16–8.09 (m, 3H, quinoxaline-**H**), 7.98–7.91 (d, *J* = 12.0 Hz, 2H, 1-naphthalene-**H**), 7.65–7.43 (t, *J* = 6.0 Hz, 2H, 1-naphthalene-**H**), 7.28 (d, *J* = 6.0 Hz, 1-naphthalene-**H**). ^13^C NMR (150 MHz, *d*-DMSO): δ_C_ (ppm): 161.0, 158.2, 147.7, 142.6, 141.4, 139.7, 134.1, 133.3, 132.8, 132.1, 131.8, 131.7, 130.3, 130.0, 129.8, 129.1, 128.8, 127.7, 127.6, 123.5, 121.2, 118.9, 108.6,79.1. HRMS (ESI+) *m*/*z* Calcd for C_24_H_16_N_4_O_2_ [M + H]^+^ 393.1112, found: 393.1350.

### 3.4. Antitumor Activity Assays

Using cisplatin as a positive control, the in vitro antitumor activity of the target compounds was determined by MTT assays [8]. A549 cells were cultured in RPMI 1640 medium at 37 °C with 5% CO_2_ and 95% air, supplemented with 10% (*v*/*v*) bovine calf serum and 80 μg/mL gentamicin. Hela cells were cultured in DMEM (high glucose) medium under the same condition as above. The logarithmic growth phase cells were inoculated in a 96-well culture plate at a concentration containing the medium at the cell density is 5 × 10^3^ cells per well. After 24 h of culture, the cells were attached to the well, and the old culture medium was discarded. The compounds to be tested were dissolved in DMSO, and the final concentration was diluted in the culture medium. The complete culture medium containing different concentrations of compounds were replaced, six concentration gradients were set (2.5 to 100 μmol/L). Each concentration had 3 complex holes, the zeroing group containing only the culture medium. After treatment for 48 h, 20 μL MTT (5 mg/mL) was added to each well, and the culture was continued for 4 h. The medium was centrifuged at 3000 rpm for 10 min, and 150 μL DMSO was added. The absorbance was measured at 570 nm by a microplate analyzer, and the IC50 value was calculated.

### 3.5. Statistical Analysis

Data were presented as from at least three independent experiments. Results were reported as mean ± SD and analyzed by Student’s *t* test. Results were considered significant when a *p* value < 0.05 was obtained.

## 4. Conclusions

PCA intermediates were synthesized from aniline and 2-bromo-3-nitrobenzoic acid. Then, PCA was esterified, and acylhydrazide was synthesized. Finally, 10 new hydrazine-1-carboxylic acid compounds containing aldehyde groups were obtained by the condensation reaction of PCA with hydrazine-containing compounds. Their structures were characterized by ^1^H and ^13^C NMR spectroscopy, MS and single crystal X-ray diffraction. The in vitro antitumor activity of the target compounds (HeLa and A549) was tested by MTT assays. The results showed that the compounds exhibited good cytotoxic activity, particularly **3d**, which has potential to be used for further studies.

## Data Availability

No new data were created or analyzed in this study. Data sharing is not applicable to this article.

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
