# Peer review of "Synthesis, Crystal Structure and Bioactivity of Phenazine-1-carboxylic Acylhydrazone Derivatives"

_molecules, 2021, doi:10.3390/molecules26175320_

Round 1

Reviewer 1 Report

NingWu, Qilong Zhang and co-workers have nicely described the synthesis and crystal structures of Phena-zine-1-carboxylic Acid Acylhydrazone Derivatives and their anti-tumor activity. Although the results are promising from scientific perspective, the description need to be elaborated and the characterization section need major corrections. This article can be considered for publications, however following major comments should to be addressed.

  1. The compound should be “3b” in the figure caption of Figure 1. Authors need to incorporate the hydrogen bonding distances in Figure 1.
  2. They also should include a comparison table indicating all hydrogen bonding distances in case of all nine crystal structures. This is crucial from the point of chelation and interaction with metal ions (iron/ferrous ions) correlated with anticancer activity.
  3. Did they use recrystallization to get the single crystals for all nine derivatives by using same method with chloroform and ethanol? They should clearly mention the crystallization techniques.
  4. In manuscript section 3.3.3, the characterization of compound 3a, 1H NMR shows the 17 protons while the compound 3a have 15 protons. They need to correct it.
  5. The interpretation and assignment of signals in case of compound 3d bearing aliphatic chain appended piperidine unit is incorrect in case of 1H as well as 13C NMR. In 1H NMR spectrum, there are also additional signals around 1.5, 2.5 and 4 ppm in figure S4 which should correspond to aliphatic protons. Similarly, in 13C there are some additional signals near 25 ppm in Figure S13 which correlated to piperidine carbons. Authors need to be more careful and should correct those.
  6. They also need to provide the ESI-MS spectra from the range of 460-480 Dalton in Figure S23. The characterization of 3d is crucial since they don’t have crystal structure and this showed the best anti-tumor activity among others tested.
  7. The legibility of Figures S4-S6 in supporting information need be improved.

Reviewer 2 Report

The article “Synthesis, Crystal Structure and Bioactivity of Phena-zine-1- carboxylic Acid Acylhydrazone Derivative” by Shouting Wu et al., describes the synthesis of 10 phenazine-1-carboxylic piperazine derivatives and their chemical characterization. They also evaluate the cytotoxic activity in two cancer cell lines (Hela and A549). Despite the interesting topic, the data herein presented are a bit confusing, often lacking the proper controls. It may have a potential to publish in Molecules after appropriate revisions. In particular:

  1. An extensive editing of English language and style is required. Some of the typos have been highlighted in the attached PDF (e.g., some hyphens should be deleted, scientific names should be in italics, etc.).
  2. The introduction must be carefully revised, as some references do not correspond with the description in the text. E.g., in line 55, the reference 24 (24. Palchykovska, L.G.; Vasylchenko, O.V.; Platonov, M.O.; Kostina, V.G.; Babkina, M.M.; Tarasov, O.A. Evaluation of antibacterial and antiviral activity of n-arylamides of 9-methyl and 9-methoxyphenazine-1-carboxylic acids – inhibitors of the phage t7 model transctiption. Biopolymers & Cell, 2012, 28(6), 477-485.) is about antibacterial activity, but not anti-leukemia.
  3. In the Results and Discussion section, relevant information is lacking:
    • ODN should be defined (line 95).
    • It should be specified which statistical significance is presented (line 103).
    • Concentration of cisplatin.
  4. It would be interesting to also perform the cytotoxic activity of these compounds in a non-cancer human cell line, to evaluate their selectivity.
  5. The discussion must be improved. Please indicate possible relationship between the chemical structure modifications and the cytotoxic activity of the compounds. Please, compare the obtained results with previous studies of other derivatives.
  6. In the Material and Methods section, it should be specified the percentage of FBS, the number of cells per well (100 L per well is not possible), the range of concentrations, and the statistical analysis.

Reviewer 3 Report

The manuscript by Wu et al. (molecules-1334424) deals with compounds with a possbile anticancer activity, and already in the title their "Crystal Structure" is announced; later (at the bottom of p. 2) there's this sentence: "In all crystal structures, the hydrogen on the amide nitrogen atom and the nitrogen atom on the phenazine ring form intramolecular N-H...N hydrogen bonds." However, I can't see actual coordinates in the supporting materials, and I can't see any info about a submission to the CCDC, either! Last week I wondered if the authors perhaps omitted to include the files, so on Friday, Jul 30, I contacted the editorial office by email and suggested to ask the authors for clarification. I haven't obtained any new information since then, so I reject the manuscript and encourage the authors to resubmit once they provide CIF files of their structures.

Round 2

Reviewer 1 Report

NingWu, Qilong Zhang and co-workers have tried to answered all the queries asked by the reviewers, however the interpretation of characterization data of compound 3d need to be corrected. These corrections are indispensable before acceptance for publication.

  1. The compound 3d has 10 aromatic protons, 1 imine, 1 amine, 1 aromatic hydroxyl and 14 aliphatic protons. But the 1H NMR in Figure S4 showed only around 8.7 (5.5+3.2) aliphatic protons, there are additional signals near 2.4 ppm (correspond to 4H, NCH2 in piperidine and 2H, NCH2-CH2O) which need to integrated and labeled. The signal at 4.12 (t) correspond to OCH2 protons (2H) instead of three. There are two signals near 1.49 (4H) and 1.40 ppm (2H). Additionally expanded spectra of each aromatic and aliphatic section should be included in supporting information along with all signals should be assigned accordingly in manuscript ans well as in supporting.
  2. Furthermore, the 13C NMR spectra signals peak position for 3d don’t correlate with the spectrum in Figure S13. For instance, first and last peak position mentioned in manuscript are 162.0 and 24.4 while the spectrum shows 161.4 and 23.8 ppm; thus these must be corrected.

Reviewer 2 Report

The manuscript “Synthesis, Crystal Structure and Bioactivity of Phena-zine-1- carboxylic Acid Acylhydrazone Derivative” by Shouting Wu et al., has been improved after revision.

However, some points should be corrected.

1.- Some of the typos that were highligted by this reviewer have not been corrected. E.g. Pseudomonads (line 47); Phena-zine (title); line 388, etc. 

2.- The statistical analysis should be more explained. Which parameters are compared? Does cisplatin exert a significant efect?

3.- The discussion section still could be improved. Authors can compare results with similar compounds in other cell lines.

Reviewer 3 Report

The manuscript by Wu et al. (molecules-1334424) now contains structure files, so in principle is publishable in Molecules. I'm glad to confirm the data for "3j" structure, which I randomly picked, are consistent between 3j.cif file and Table S4. I think the CCDC submission numbers have to be given in Data Availability Statement. I'm sorry to see "Phena-zine" in the title (the hyphen??) and "Thiazolyl Blue Tetrazolium Bromide" in the abstract (the caps??), but I presume these and other issues can be resolved during a revision of the text that summarizes a lot of work done (I can also see changes based on previous numerous comments of the other reviewers).
